# A dynamic mode of mitotic bookmarking by transcription factors

**Sheila S Teves[1], Luye An[1], Anders S Hansen[1,2], Liangqi Xie[1,2], Xavier Darzacq[1,2]\*, Robert Tjian[1,2,3]\***

[1]Department of Molecular and Cell Biology, University of California, Berkeley, United States; [2]CIRM Center of Excellence, University of California, Berkeley, Berkeley, United States; [3]Howard Hughes Medical Institute, University of California, Berkeley, Berkeley, United States

**Abstract** During mitosis, transcription is shut off, chromatin condenses, and most transcription factors (TFs) are reported to be excluded from chromosomes. How do daughter cells re-establish the original transcription program? Recent discoveries that a select set of TFs remain bound on mitotic chromosomes suggest a potential mechanism for maintaining transcriptional programs through the cell cycle termed mitotic bookmarking. Here we report instead that many TFs remain associated with chromosomes in mouse embryonic stem cells, and that the exclusion previously described is largely a fixation artifact. In particular, most TFs we tested are significantly enriched on mitotic chromosomes. Studies with Sox2 reveal that this mitotic interaction is more dynamic than in interphase and is facilitated by both DNA binding and nuclear import. Furthermore, this dynamic mode results from lack of transcriptional activation rather than decreased accessibility of underlying DNA sequences in mitosis. The nature of the cross-linking artifact prompts careful re-examination of the role of TFs in mitotic bookmarking.

\*For correspondence: darzacq@ berkeley.edu (XD); jmlim@ berkeley.edu (RT)

## Introduction

A key component of cell identity is the epigenetic maintenance of cell-type specific transcription programs. However, this maintenance is challenged during each cell cycle. In mitosis, the global transcriptional machinery is inactivated via a cell cycle-dependent phosphorylation cascade (*Prescott and Bender, 1962*; *Rhind and Russell, 2012*; *Taylor, 1960*). Interphase chromatin organizes into highly condensed mitotic chromosomes (*Koshland and Strunnikov, 1995*), and the nuclear envelope is disassembled (*Terasaki et al., 2001*). Furthermore, most transcription factors (TFs) have been shown to be excluded from mitotic chromosomes (*Gottesfeld and Forbes, 1997*; *John and Workman, 1998*; *Martínez-Balbás et al., 1995*; *Rizkallah and Hurt, 2009*), leading to the conclusion that mitotic chromosomes may be inaccessible to DNA binding and exclude most TFs. Following mitosis, how then do the new daughter cells faithfully re-establish the cell-type specific transcription program?

Several mechanisms have been proposed to play important roles in re-establishing transcription following mitosis (*Lodhi et al., 2016*). These include maintenance of DNA methylation patterns for heritable silencing and the propagation of histone modifications, although a model to account for targeted modifications in the absence of TFs has been elusive. Moreover, there are indications that DNA methylation and histone modifications are not sufficient to maintain transcription profiles through the cell cycle. For example, DNase I hypersensitive sites on the human *hsp70* locus were shown to be maintained in mitotic chromosomes (*Martínez-Balbás et al., 1995*) implying the presence of a 'bookmarker' to keep the region accessible to nuclease digestion. Similarly, the transcription start sites (TSSs) of certain genes scheduled for reactivation following mitosis were shown to

**eLife digest** A kidney cell functions differently from a skin cell despite the fact that all the cells in one organism share the same DNA. This is because not all of the genes encoded within the DNA are active in the cells. Instead, cells can turn on just those genes that are specific to how that cell type works. One way that cells can regulate their genes is by using proteins called transcription factors that can bind to DNA to turn nearby genes on and off.

When cells divide to form new cells, the DNA is condensed and gene activity is turned off. However, each dividing cell also has to 'remember' the program of genes that specifies its identity. After division, how do the cells know which genes to turn on and which ones to keep off?

It was thought that the transcription factors attached to the DNA were all detached from it during cell division. Through studies in mouse embryonic stem cells, Teves et al. now show that this finding is largely an artifact of the methods used to study the process. In fact, many transcription factors still bind to and interact with DNA during cell division. This provides an efficient way for the newly formed cells to quickly reset to the pattern of gene activity appropriate for their cell type.

Having found that many key transcription factors are still bound to DNA during cell division, the next challenge is to find out what role this binding plays in allowing cells to 'remember' their identity.

remain sensitive to permanganate oxidation in mitosis, suggesting a conformationally privileged structure at the TSSs of these genes (*Michelotti et al., 1997*). It was thus proposed that some unknown factors must escape the exclusion from mitotic chromosomes and bookmark these regions, yet none have been shown to remain bound on chromosomes. It was therefore a significant step in resolving this conundrum when HSF2 was shown to bind at the *hsp70i* locus during mitosis (*Xing et al., 2005*). Since then, and coincident with the advent of live-cell microscopy, a few other TFs have been discovered to associate with mitotic chromosomes (*Caravaca et al., 2013*; *Kadauke et al., 2012*; *Lodhi et al., 2016*), beginning a re-emergence of an appreciation for TFs in propagating transcription programs through mitosis. For instance, GATA1, a major regulator of the erythroid lineage, has previously been reported to be excluded from mitotic chromosomes by immu-nofluorescence (*Xin et al., 2007*). Subsequently, the Blobel group has shown, by live-cell imaging and chromatin immunoprecipitation analysis, that GATA1 actually remained bound on its target regions during mitosis (*Kadauke et al., 2012*). TFs such as GATA1 seem to act as the elusive 'book-mark' that maintain chromatin architecture at regulatory regions, and thus have been termed mitotic bookmarkers. Despite several recent examples of TFs that have been identified as potential mitotic bookmarkers (*Lodhi et al., 2016*), these have generally been regarded as special cases while most of the literature document robust eviction of TFs from chromosomes during mitosis.

Using a combination of in vitro biochemical assays, genome editing, and fixed versus live-cell imaging, we report that contrary to decades of published literature, most TFs we tested remain associated with mitotic chromosomes. The widely observed exclusion of TFs from mitotic chromo-somes is due primarily to a formaldehyde-based cross-linking artifact. Sox2, for example, appears excluded from chromosomes after chemical fixation, but is highly enriched on mitotic chromosomes as determined by live-cell imaging. This enrichment of TFs at mitotic chromosomes is facilitated by both the DNA binding domain of Sox2 and by active nuclear import. Using orthogonal imaging approaches such as single particle tracking and fluorescence recovery after photobleaching, we show that Sox2 binds dynamically to mitotic chromosomes, and that this dynamic behavior relates to the absence of transcriptional activation rather than a global inaccessibility of DNA in condensed chromosomes. These findings led us to investigate how chemical fixation may alter the localization of TFs in mitotic cells. We present a model for the mechanistic action of formaldehyde-based cross-linkers on transcription factor localization, and consider the overarching implications of this cell fixa-tion artifact on interpreting experiments designed to study many biological processes and particu-larly transcriptional bookmarking.

## Results

### Many transcription factors associate with mitotic chromosomes

We initially hypothesized that Sox2, one of the key pluripotency TFs in embryonic stem cells, may function as a mitotic bookmarker to maintain the ES cell state. To examine whether Sox2 binds to mitotic chromosomes, we synchronized cells at various stages of the cell cycle and obtained about 95% pure mitotic population. (*Figure 1—figure supplement 1*). We then performed biochemical fractionation to assess the chromatin-bound fraction on the asynchronous (A), mitotic (M), G2- and S- phase cells (*Figure 1—figure supplement 2*). We detected Sox2 on chromatin fractions from synchronized populations, including mitotic cells (*Figure 1A*), providing initial evidence that Sox2 may associate with mitotic chromosomes. Similarly, TBP fractionated with mitotic chromosomes whereas Pol II did not (*Figure 1—figure supplement 2*). To biochemically assess the strength of this association, we performed salt fractionation on asynchronous and mitotic cells (*Figure 1—figure supplement 2*). Nuclear transcription factors elute from chromatin at the salt concentration that overcomes their binding strength to DNA. In the asynchronous population, the majority of Sox2 fractionated at high salt and in the micrococcal nuclease-digested chromatin, suggesting a strong interaction of Sox2 with chromatin (*Figure 1B*). In contrast, Oct4 fractionated with much lower salt concentrations (*Figure 1B*), consistent with its more dynamic association with chromatin (*Chen et al., 2014*). Sox2 displayed a similar salt fractionation profile in synchronized mitotic cells, albeit with a somewhat reduced signal in the digested chromatin (*Figure 1B*). TBP also showed a strong association with mitotic chromosomes whereas Pol II was primarily cytoplasmic (*Figure 1—figure supplement 2*). Taken together with the biochemical fractionation assay, these data suggest that Sox2 likely associates with mitotic chromosomes in a manner that is qualitatively weaker than its association with interphase chromatin.

To visualize Sox2 in mitotic cells, we performed immunofluorescence analysis of Sox2 in standard formaldehyde-fixed mouse ES cells stably expressing H2B-GFP. In contrast to our biochemical data, we found that Sox2 is largely excluded from mitotic chromosomes (*Figure 1C*). However, when we over-expressed Halo-tagged Sox2 (Halo-Sox2 OE) in mouse ES cells stably expressing H2B-GFP and imaged mitotic cells under live-cell conditions, we observed that Halo-Sox2 is highly enriched on mitotic chromosomes (*Figure 1D*). To resolve the discrepancy between the fixed immunofluorescence data versus the over-expressed live-cell imaging, we sought to endogenously knock-in the HaloTag at the Sox2 locus using the CRISPR/Cas9 system such that all endogenous Sox2 molecules could be visualized. We obtained three independent clones that were homozygously tagged (Halo-Sox2 KI), and confirmed that the tagging has no detectable effect on Sox2 function as Halo-Sox2 KI ES cells maintain pluripotency (*Figure 1—figure supplement 3*). Imaging under live-cell conditions, we confirmed that Halo-Sox2 KI is indeed highly enriched on mitotic chromosomes (*Figure 1D*). Furthermore, using time-lapse microscopy, we detected strong enrichment of Halo-Sox2 KI to chromosomes throughout all stages of mitosis (*Figure 1E*). To validate that the signal is indeed from the endogenous Sox2 and not an artifact of the knock-in, we first labeled the Halo-Sox2 KI in live cells followed by fixation and standard immunofluorescence using α-Sox2. The signal for the HaloTag and α-Sox2 co-localized, confirming that the HaloTag is fused correctly with Sox2 (*Figure 1F*). Surprisingly, both the HaloTag and the α-Sox2 signals in these fixed cell preparations were found to be excluded from mitotic chromosomes as marked by H2B-GFP (*Figure 1F*) in contrast to the live-cell imaging results. We performed the same fixation experiment on ES cells over-expressing Halo-Sox2 and observed the same chromosome exclusion phenomenon typically reported for TFs (*Figure 1F*). We quantified the change in enrichment on mitotic chromosomes by taking the $\log_2$ ratio of the mean intensity on chromosomes over the whole cell intensity, using the H2B-GFP as a mask for chromosomes (*Figure 1G*). Positive values correspond to enrichment on mitotic chromosomes whereas negative values signify exclusion from chromosomes. With this metric, both Halo-Sox2 OE and Halo-Sox2 KI switched from positive enrichment under live-cell imaging conditions to exclusion after fixation (*Figure 1H*), suggesting that fixation by paraformaldehyde results in the exclusion of Sox2 from mitotic chromosomes. This result is reminiscent of a previous study showing that HMGB proteins become excluded from mitotic chromosomes after fixation (*Pallier et al., 2003*). The authors argue that formaldehyde may uniquely alter the structure of HMGB proteins and thus result in exclusion, which may be consistent with Sox2 as it contains an HMG domain. However, because most of the evidence for TF exclusion derives from fixed cell immunofluorescence analysis, we wondered how

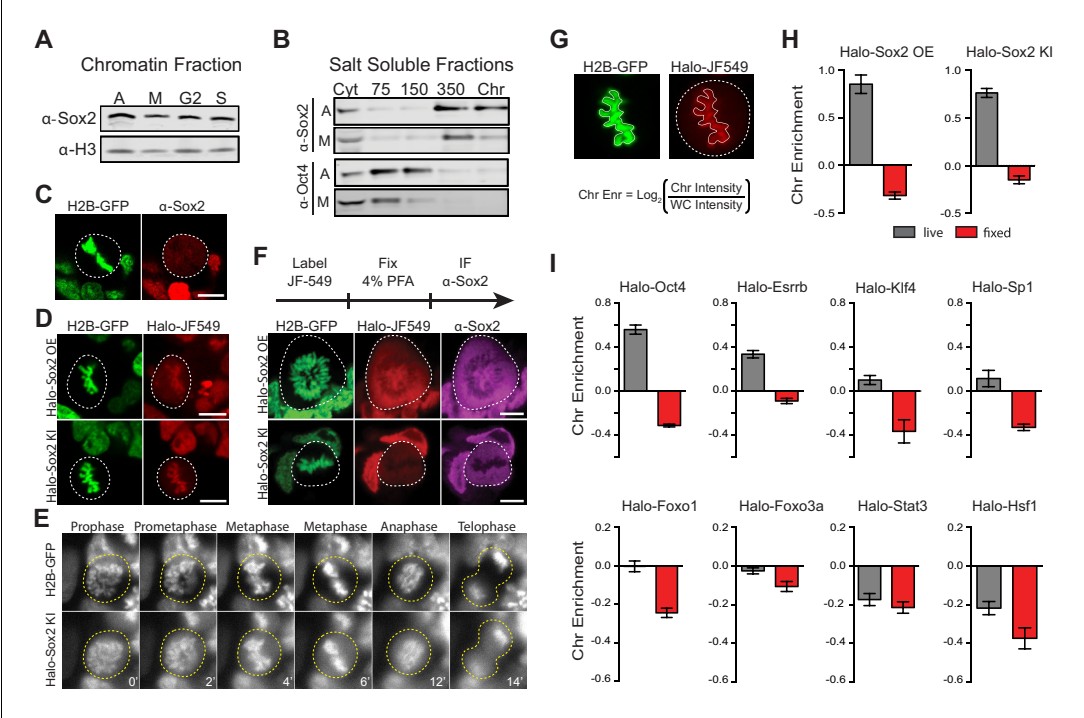

**Figure 1.** Transcription factors are not excluded from mitotic chromosomes. (**A**) Biochemical fractionation of asynchronous (A) mouse ES cells and synchronized populations at mitosis (M), G2, and S phases was performed to isolate the chromatin-associated fraction. Sox2 and H3 were detected by Western blot analysis. (**B**) Salt fractionation of asynchronous (A) and mitotic (M) mouse ES cells. Sox2 and Oct4 were detected by Western blot analysis. Cyt, Cytoplasmic fraction. Chr, Chromatin fraction. (**C**) Immunofluorescence with α-Sox2 of mouse ES cells stably expressing H2B-GFP using confocal microscopy showing exclusion of Sox2 from mitotic chromosomes (**D**) Live-cell imaging using confocal microscopy of mouse ES cells stably expressing H2B-GFP with overexpressed Halo-Sox2 (Halo-Sox2 OE, top) and endogenously tagged Halo-Sox2 (Halo-Sox2 KI, bottom) (**E**) Epi-fluorescence time-lapse imaging of mouse ES cells stably expressing H2B-GFP and endogenously-tagged Halo-Sox2 KI (**F**) Live cells with overexpressed Halo-Sox2 (top) or endogenously-tagged Halo-Sox2 (bottom) were labeled with JF549 dye and subjected to standard immunofluorescence by fixation with 4% PFA and detection with α-Sox2. (**G**) Strategy for quantifying TF chromosome enrichment. (**H**) Chromosome enrichment levels for overexpressed and endogenously-tagged Halo-Sox2. n = 40 cells (**I**). Chromosome enrichment levels for indicated Halo-tagged transcription factors. n = 40 cells. Data are represented as mean ± SEM. Scale bars, 5 µm.

The following figure supplements are available for figure 1:

**Figure supplement 1.** Synchronization of mouse ES cells.

**Figure supplement 2.** Biochemical and Salt fractionation strategies.

**Figure supplement 3.** Endogenous knock-in of HaloTag to Sox2 locus using CRISPR/Cas9.

**Figure supplement 4.** Live versus fixed images of Halo-tagged TFs in mouse ES cells.

**Figure supplement 5.** Controls for live versus fixed imaging.

general this artifact might be. We generated stable cell lines expressing a variety of Halo-tagged TFs in mouse ES cells, including Oct4, Esrrb, Klf4, Sp1, Foxo1, Foxo3a, Stat3, and Hsf1, and performed the same quantitative analysis of live versus fixed images. In all cases, the level of enrichment on mitotic chromosomes dramatically decreased after fixation, resulting in an apparent exclusion of all TFs examined (*Figure 1I* and *Figure 1—figure supplement 4*). When imaged under live-cell conditions, however, the majority of these factors exhibited varying levels on mitotic chromosomes, from highly enriched to uniform levels. This is consistent with a study showing a large number of TFs associated with mitotic chromosomes in chicken DT40 cells (*Ohta et al., 2010*). We found two TFs

that are excluded from mitotic chromosomes under live-cell conditions, Stat3 and Hsf1 (*Figure 1I*), both of which are known to require external signals to translocate into the nucleus during interphase. We have also tested other commonly used fixation methods and have varied the fused tag, and all the results suggest that chemical fixation somehow evicts TFs from mitotic chromosomes (*Figure 1— figure supplement 5*). This surprising finding suggests that a large body of literature spanning several decades of studying TFs during mitosis, from their widely accepted exclusion from mitotic chromosomes to the apparent uniqueness of certain select TFs as specialized 'mitotic bookmarkers', may have been founded in large part on a fixation artifact.

## Dissection of Sox2 interaction with mitotic chromosomes

Given the varying levels of TFs on mitotic chromosomes, we next investigated the intrinsic TF properties that might determine the association with mitotic chromosomes. Sox2, for example, is composed primarily of two domains, the N-terminal high mobility group (HMG) DNA binding domain and the C-terminal trans-activation domain (TAD) (*Figure 2A*). The HMG domain contains sequence-specific DNA binding residues as well as nuclear localization signals (NLS), whereas the TAD is integral to protein-protein interactions with Sox2 partner proteins, including Oct4 and p300 (*Cox et al., 2010*). To address which regions are important for mitotic enrichment, we expressed truncations of Sox2 as HaloTag fusions in ES cells stably expressing H2B-GFP. Halo-Sox2 HMG became highly enriched on mitotic chromosomes whereas Halo-Sox2 TAD was mostly cytoplasmic (*Figure 2B*). When we quantified the chromosome enrichment of the truncations, we observed that the HMG domain is sufficient to render mitotic chromosome localization (*Figure 2B*). We next tested whether sequence-specific binding and/or the NLS within the HMG domain is necessary for mitotic enrichment. We mutated five amino acid residues that have been shown to contact the DNA minor groove (N48A, N70A, S71A, S74A, Y112A) (*Reményi et al., 2003*), and fused this to the HaloTag (Halo-Sox2 DBD5M). Expressed in ES cells, mutations of DNA binding residues resulted in exclusion from mitotic chromosomes under live-cell imaging conditions (*Figure 2B*). A separate study has shown that mutating three distinct residues (M47G, F50G, M51G) also abolished DNA binding (*Chen et al., 2014*). Expressing this construct (Halo-Sox2 DBD3M) also showed exclusion from mitotic chromosomes (*Figure 2B*), confirming that DNA binding is necessary for mitotic enrichment. We next tested what role Sox2 NLS might play on mitotic enrichment. Sox2 contains two NLS, a bipartite signal located at residues 43–46 (VKRP) and 57–60 (QRRK), and a monopartite signal located at residues 114–117 (PRRK) (*Polakova et al., 2014*). To abolish the NLS completely, we introduced mutations at both NLS elements, specifically K44A, R45A, R58A, R59A, R115A, R116A, and K117A. Surprisingly, mutations to the NLS also resulted in exclusion from mitotic chromosomes (*Figure 2B*).

Although the Sox2 NLS is in close proximity to the DNA binding residues and as such, mutations to the NLS may affect DNA binding, some indication for the role of nuclear import in the mitotic enrichment of TFs has recently been shown. For instance, mutants of HNF1B become enriched on mitotic chromosomes following cold shock, and that this enrichment is dependent on nuclear import (*Lerner et al., 2016*). We therefore wondered if an NLS is sufficient to confer enrichment on mitotic chromosomes. We expressed a fusion of the HaloTag and the SV-40 NLS in mouse ES cells stably expressing H2B-GFP. When imaged under live conditions, we discovered that the NLS enriched the HaloTag protein in mitotic chromosomes (*Figure 2C*). In contrast, HaloTag protein alone is excluded from mitotic chromosomes (*Figure 2C*). There are two potential mechanisms that could enrich an NLS-containing protein on mitotic chromosomes. The first is that highly positive residues on NLSs, generally lysines and arginines, interact non-specifically with the negatively charged chromosomes. In the case of the SV-40 NLS, the sequence is PKKKRKV. The second potential mechanism is that the nuclear import machinery actively enriches nuclear factors on mitotic chromosomes. To distinguish between these two possibilities, we expressed the fusion of a plant-specific NLS with the HaloTag in mouse ES cells stably expressing H2B-GFP. The plant NLS contains positively charged residues (SVLGKRKFA), but is functional only in plants (*Kosugi et al., 2009*). When imaged under live-cell conditions, we observed no enrichment of Halo-Plant NLS on mitotic chromosomes (*Figure 2C*). These results suggest that both sequence-specific binding and active nuclear import likely contribute to the enrichment of TFs on mitotic chromosomes.

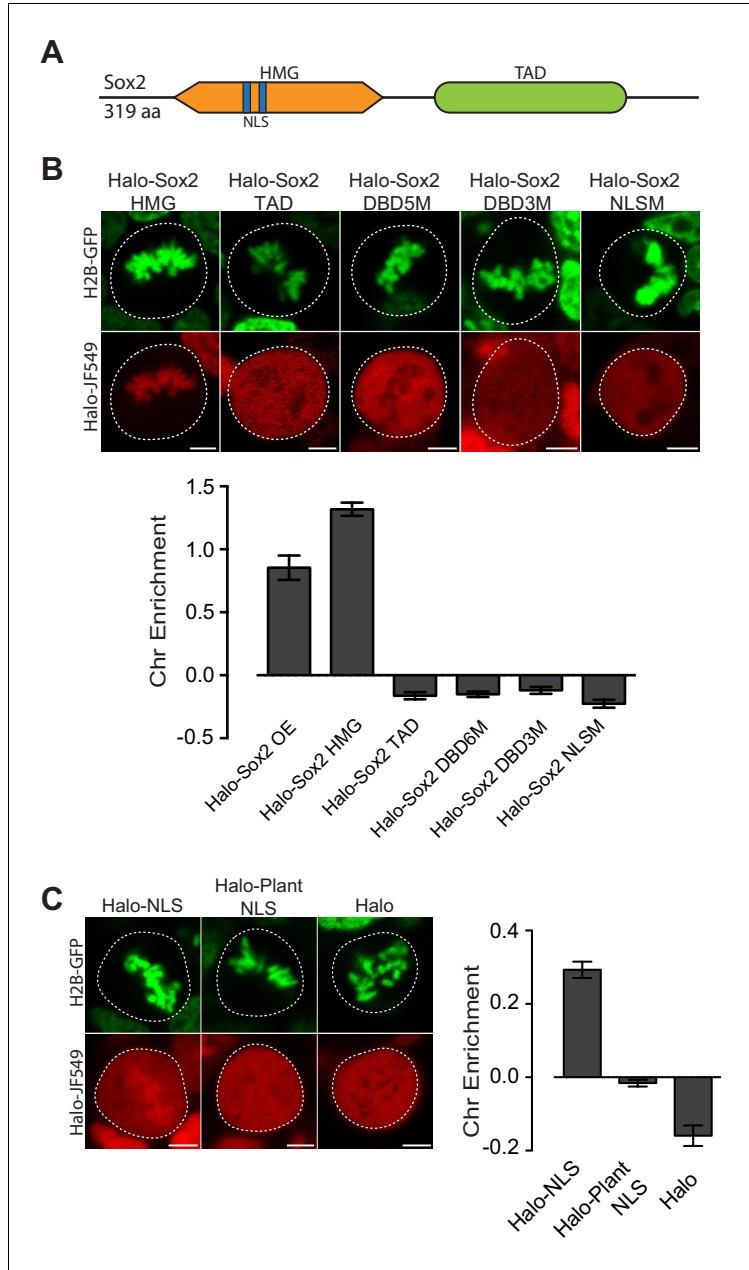

**Figure 2.** Mitotic enrichment of Sox2 requires DNA binding and nuclear import. (**A**) Schematic of Sox2 domains. HMG, High Mobility Group domain. TAD, Transactivation Domain. NLS, Nuclear Localization Signal. (**B**) Live-cell imaging of mouse ES cells stably expressing H2B-GFB and various Halo-tagged Sox2 truncations or mutations. Bottom, chromosome enrichment quantification for the various Halo-tagged Sox2 over-expressing (OE) constructs. Halo-Sox2 HMG construct is a truncation of Halo-Sox2 with the TAD region deleted. Halo-Sox2 TAD is a truncation of Halo-Sox2 with the HMG domain deleted. Halo-Sox2-DBD5M and Halo-Sox2 DBD3M are the full length Halo-Sox2 with 5 and 3 point mutations to abrogate DNA binding, respectively. Halo-Sox2 NLSM is the full length Halo-Sox2 with point mutations to abolish NLS function. n = 40 cells. (**C**) Live-cell imaging of mouse ES cells stably expressing H2B-GFP and HaloTag fused to SV40 NLS, plant-specific NLS, or by itself. Right, chromosome enrichment levels for the indicated HaloTag constructs. n = 40 cells. Data are represented as mean ± SEM. Scale bars, 5 μm.

## Sox2 interaction with mitotic chromosomes is highly dynamic

We next examined the interaction dynamics of TFs on mitotic chromosomes using Sox2 as a model through fluorescence recovery after photobleaching (FRAP) (*Figure 3A*). Quantifying intensities at the bleach spot over time shows that Halo-Sox2 recovery is faster in mitotic cells than during interphase, but significantly slower than Halo-NLS (*Figure 3B*). The average time for Halo-Sox2 to reach 90% recovery in interphase is 19.7 s, whereas 90% recovery in mitosis is 4.3 s on average (*Figure 3C*). In contrast, Halo-NLS reaches 90% recovery in 0.9 s for both interphase and mitotic cells (*Figure 3C*). These results suggest that the interaction of Sox2 with mitotic chromosomes is more dynamic than its interaction with interphase chromatin. As an orthogonal approach to FRAP, we performed single particle tracking (SPT) with long exposure times (500 ms) to determine residence times of Halo-Sox2 in mitosis relative to interphase (*Videos 1* and *2*). The long exposure times allow for a 'blurring out' of fast moving molecules while immobile, 'stably' bound molecules appear as bright diffraction-limited spots as previously reported for Sox2 (*Chen et al., 2014*). We plotted the semi-log histogram of Halo-Sox2 dwell times, the amount of time each molecule remains detected, for interphase and mitotic cells (*Figure 3D*). A two-component exponential decay model, representing specific versus non-specific binding events, was fit to the dwell time histograms as previously reported to extract the residence times of bound molecules (*Chen et al., 2014*). After correction for photobleaching, the residence time of specific Halo-Sox2 binding events during mitosis is 54% relative to interphase (*Figure 3D* inset). The decreased residence times for specific Sox2 interactions in mitosis relative to interphase is comparable with the faster FRAP recovery of Sox2 during mitosis. We next investigated whether there is also a change in the total fraction of molecules that are bound versus freely diffusing by performing SPT at faster imaging frequencies (223 Hz) (*Videos 3* and *4*). For each tracked molecule, we measured the displacement of individual molecules between frames (jump length), and plotted the histogram of jump lengths for interphase and mitotic cells (*Figure 3E*). We then fitted a 2-state model to the probability distribution of jump lengths as previously reported (*Mazza et al., 2012*) to extract the fraction of molecules that are bound versus freely diffusing. We performed the same analysis for H2B-Halo and Halo-NLS to distinguish bound (short jump lengths) and freely diffusing (large jump lengths) states, respectively (*Figure 3—figure supplement 1*). In interphase, 30.9% of Halo-Sox2 molecules are bound, whereas 18.3% of Halo-Sox2 molecules are bound in mitosis (*Figure 3E* inset), an almost two-fold decrease in bound population. This total bound fraction includes both short- and long-lived binding events. Taken together with the slow tracking SPT analysis, these data suggest that there are fewer Sox2 molecules bound during mitosis, and that these bound molecules experience a faster off-rate in mitosis.

Why is Sox2 interaction with mitotic chromosomes more dynamic than with interphase chromatin? By examining truncations of Sox2, we have discovered that the HMG domain of Sox2 is important for mitotic enrichment, but what are the contributions of the TAD on the dynamics of this interaction? To answer this question, we performed FRAP analysis on mouse ES cells expressing Halo-Sox2 HMG in interphase and mitotic cells. Compared to full length Halo-Sox2, the Halo-Sox2 HMG showed faster FRAP recovery both in interphase and mitosis (*Figure 3B–C*). Quantification of fluorescence intensities over time at the bleach spot shows that Halo-Sox2 HMG in interphase and mitosis resemble that of Halo-Sox2 in mitosis (*Figure 3C*). Indeed, the average time for Halo-Sox2 HMG to reach 90% recovery in both interphase and mitosis is 6.6 s, comparable to Halo-Sox2 in mitosis (4.3 s) (*Figure 3C*). These results suggest that the TAD plays a significant role in stabilizing Sox2 interactions with interphase chromatin. The TAD region participates in interactions with partner proteins, including Oct4 and p300, to activate transcription (*Cox et al., 2010*). Given these results, one possible model is that the HMG domain of Sox2 allows for initial contact with target DNA sites, and transcriptional activation by the TAD stabilizes Sox2 at the main target sites via protein-protein interactions with partner factors, perhaps in the assembled pre-initiation complex (PIC). During mitosis, transcription is shut off globally, and therefore, the full length Sox2 now interacts with mitotic chromosomes primarily through its DNA binding domain, rendering the TAD inactive.

## Global DNA accessibility is unaltered during mitosis

The lack of transcriptional activation during mitosis may contribute to the more dynamic interaction of Sox2 with mitotic chromosomes, but another possibility is that the highly condensed nature of mitotic chromosomes also decreases the ability of Sox2 to access its binding sites. To test if mitotic

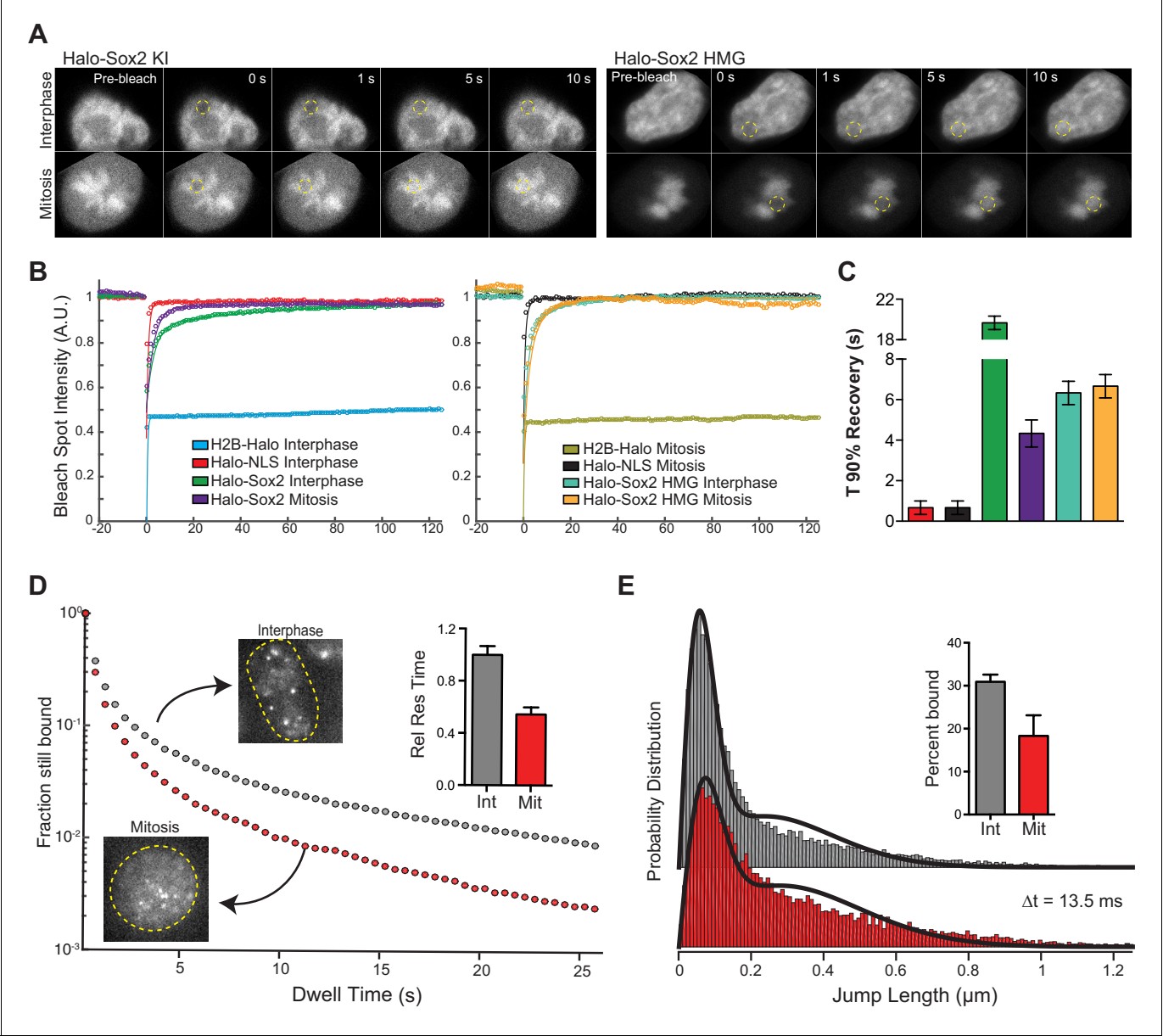

**Figure 3.** Sox2 interaction with mitotic chromosomes is highly dynamic. (A) FRAP analysis of HaloSox2 KI and HaloSox2 HMG cells for interphase and mitosis. (B) Quantification of fluorescence recovery at the bleach spot for the indicated Halo-tagged construct in interphase and mitosis. n = 30 cells. (C) From (B), the average time to reach 90% recovery for the indicated (color-coded) Halo-tagged construct. (D) Dwell time histogram of the fraction of endogenously-tagged Halo-Sox2 molecules remaining bound for interphase (gray) and mitotic (red) cells. Representative images are shown. Inset, quantification of the relative Sox2 residence time as percentage of interphase cells. n = 30 cells. (E) Jump length histogram for three consecutive images (Δt = 13.5 ms) of the endogenously-tagged Halo-Sox2 molecules for interphase (gray) and mitotic (red) cells. A 2-state model is used to fit the histogram (solid line), and the fraction bound is calculated (inset). n = 24 cells. Data are represented as mean ± SEM.

The following figure supplement is available for figure 3:

**Figure supplement 1.** Controls for single particle tracking experiments.

chromosomes have decreased accessibility, we utilized Assay for Transposase-Accessible Chromatin using sequencing (ATAC-seq) analysis where the integration of sequencing-compatible adapters by the Tn5 transposase is directly correlated with the accessibility of genomic regions (*Buenrostro et al., 2013*). We performed ATAC-seq in asynchronous cells as well as in Nocodazole-

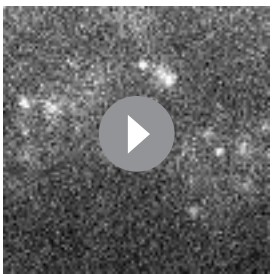

**Video 1.** SPT for residence time analysis of Halo-Sox2 KI cells in interphase. Related to *Figure 3*. Imaging immobile Halo-Sox2 molecules in interphase ES cells at 2 Hz. Movie fps = 20. one pixel = 160 nm.

synchronized mitotic cells in two biological replicates each. Synchronized mitotic cells remain mitotic throughout the ATAC-seq protocol, and biological replicates for both asynchronous and mitotic cells were highly correlated with each other and were therefore combined in subsequent analyses (*Figure 4—figure supplement 1*). The length distribution profile of sequenced fragments from asynchronous cell populations and synchronized mitotic cells revealed near superimposable patterns in Tn5 integration (*Figure 4A*), including the 10 bp periodicity patterns that mark DNA helical pitch (*Figure 4A* inset). These nearly identical integration patterns and sequencing read counts (*Figure 4—figure supplement 1*) suggest that Tn5 transposase may access mitotic chromosomes as equally well as interphase chromatin. Indeed, when we mapped the sequenced reads and visualized them on the genome, we observed nearly identical patterns and intensities of peaks (*Figure 4B*). We further parsed the reads computationally based on length, with short reads (under 100 bp) representing sub-nucleosomal DNA regions, and reads between 180–247 bp representing mono-nucleosome sized fragments (*Figure 4A* inset). We then mapped these size classes separately and obtained distinct genomic profiles (*Figure 4C*). Similar to the total mapped reads, there is little qualitative difference in peak patterns and intensities between asynchronous and mitotic cells when we analyze both the short reads and the mono-nucleosome sized fragments (*Figure 4C*). To quantitatively assess the level of concordance between asynchronous and mitotic cells, we called peaks individually for both samples and combined unique peaks. We then measured the intensities of each called peak for both asynchronous and mitotic samples and plotted the values as a log-scale scatter plot heatmap (*Figure 4D*). This analysis shows a near perfect symmetry along the diagonal, suggesting that the peak intensities between asynchronous and mitotic cells are concordant. Indeed, linear regression analysis shows the slope of the fit at 0.965, with an $R^2$ value of 0.991 (*Figure 4D*). Performing the same analysis on the short reads (*Figure 4E*) and mono-nucleosome sized fragments (*Figure 4F*) also yields symmetrical patterns along the diagonal, with linear fits close to one. These quantitative analyses indicate that mitotic chromosomes are accessed by the Tn5 transposase in nearly the same manner as it accesses interphase chromatin, suggesting that the massive condensation of mitotic chromosomes has little effect on DNA accessibility.

## DNA accessibility of Sox2 binding sites is maintained in mitosis

Global analysis of DNA accessibility shows no marked difference between interphase chromatin and mitotic chromosomes, but what about at specific TF binding sites such as those of Sox2? We exam-

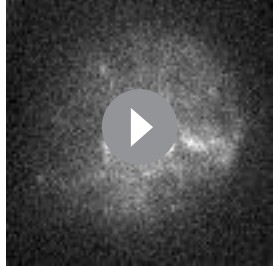

**Video 2.** SPT for residence time analysis of Halo-Sox2 KI cells in mitosis. Related to *Figure 3*. Imaging immobile Halo-Sox2 molecules in mitotic ES cells at 2 Hz. Movie fps = 20. one pixel = 160 nm.

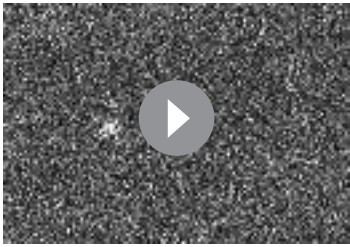

**Video 3.** SPT for fraction bound measurements of Halo-Sox2 KI cells in interphase. Related to *Figure 3*. Imaging fast Halo-Sox2 molecules in interphase ES cells at 223 Hz. Movie fps = 20. one pixel = 160 nm.

ined total ATAC-seq reads as well as short and mono-nucleosome sized fragments on the well-characterized Oct4 distal enhancer (DE) that contains binding sites for Sox2 (*Figure 5A*). We observed a prominent peak in short reads centered at the DE that is flanked by well-defined nucleosomes in asynchronous cells. Importantly, the peak of ATAC-seq reads in between the two well-positioned nucleosomes is maintained in mitosis (*Figure 5A*). To assess accessibility for all bound Sox2 sites, we collated significantly bound Sox2 regions from published ChIP-seq experiments for Sox2 in mouse ES cells (*Chen et al., 2008*), and analyzed ATAC-seq data at and surrounding these binding sites in

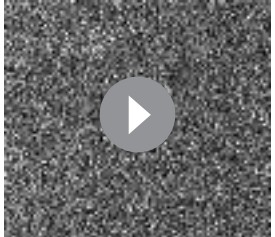

**Video 4.** SPT for fraction bound measurements of Halo-Sox2 KI cells in mitosis. Related to *Figure 3*. Imaging fast Halo-Sox2 molecules in mitotic ES cells at 223 Hz. Movie fps = 20. one pixel = 160 nm.

asynchronous and mitotic samples. The heatmap of ATAC-seq signal on a 4 kb region centered at the Sox2 peaks shows a prominent region of accessibility at the peak center in asynchronous cells that is maintained albeit at somewhat reduced levels in mitotic cells (*Figure 5B*, left). Averaging the integration density at all Sox2 bound sites shows that mitosis integration density is roughly 60% of the integration density in asynchronous cells (*Figure 5B*, right). This significant decrease in integration density at Sox2 bound regions may reflect the reduced residence time of Sox2 at these sites during mitosis (*Figure 2D*), and is consistent with a previous study showing decreased accessibility at enhancer regions (*Hsiung et al., 2015*). However, this result may not reflect the ability of Sox2 to physically access the full repertoire of binding regions during mitosis. To assess whether Sox2 can sample its binding sites during mitosis, we collated regions of the genome containing the Sox2 binding motif. We averaged the integration density at single base resolution in an 80 bp region centered at the Sox2 motif to precisely define the footprint of Sox2 in asynchronous and mitotic cells (*Figure 5C*). This footprinting analysis shows nearly identical integration patterns and footprinting depth for asynchronous and mitotic cells that is absent in matching random genomic sites, suggesting that Sox2 can sample binding sites as equally well in mitotic as in asynchronous cells. This interpretation is consistent with the similar binding dynamics of the Sox2 HMG domain in interphase and in mitosis (*Figure 3B*), lending further support for the model that Sox2 scans its binding sites relatively quickly through its DNA binding domain and is stabilized through the TAD when transcription is activated.

## Potential mechanism of chemical fixation artifact

Our surprising discovery of a formaldehyde fixation artifact prompted us to examine a potential mechanism for the artificial eviction of TFs. To capture this artifact in action, we performed time-lapse two-color imaging at 2 Hz of mitotic Halo-Sox2 KI cells stably expressing H2B-GFP as we added 1% paraformaldehyde (PFA) to the cells (*Figure 6A* and *Video 5*). At 2 s before 1% PFA is added, Halo-Sox2 is highly concentrated on mitotic chromosomes as marked by H2B-GFP. Within 10 s after PFA addition, Halo-Sox2 levels at the chromosomes are visibly reduced, and becomes almost indistinguishable from the cytoplasmic signal by 60 s after PFA addition (*Figure 6A*). These results point to a robust effect of PFA in inducing the apparent eviction of TFs from mitotic chromosomes. A potential mechanistic model through which PFA could produce such an artifact is as follows (*Figure 7*). As PFA molecules cross the cell membrane, they rapidly cross-link to the nearest protein available. This would result in a steep gradient of cross-linking that moves inward as more and more PFA molecules cross the membrane. The non-uniform rate and directionality of cross-linking would likely deplete the cytoplasmic pool of TFs that could associate with chromosomes. Furthermore, the initial cross-linking of the cytoplasmic pool would result in an effective decrease in $k_{on}$ of TFs on mitotic chromosomes. We note that recent studies have shown, using live-cell single molecule tracking and FRAP experiments, that most TFs have a residence time of under 20 s and are thus quite dynamic even during interphase (*Chen et al., 2014*; *Elf et al., 2007*; *Gebhardt et al., 2013*; *Mueller et al., 2008*; *Swinstead et al., 2016*). At least in the case of Sox2, residence time is also decreased in mitosis. This increase in effective $k_{off}$ of TFs superimposed on this moving gradient of cross-linking would result in the apparent exclusion of TFs from mitotic chromosomes. This model

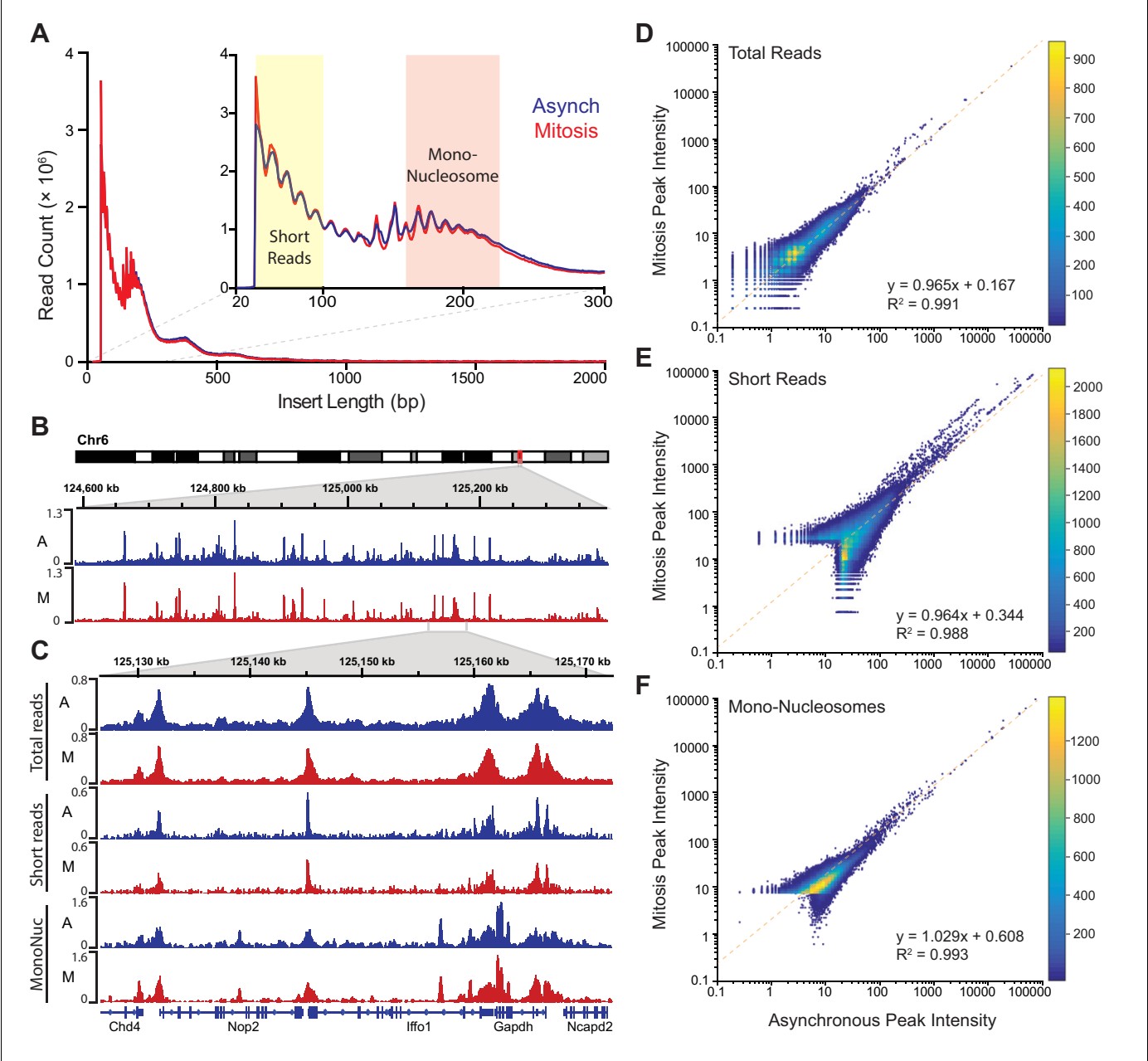

**Figure 4.** Global accessibility is maintained in mitotic chromosomes. (**A**) Fragment length distribution of ATAC-seq reads for asynchronous (blue) and mitotic (red) cells. Inset, magnification of fragment length distribution under 300 bp showing size cut-offs for short reads (under 100 bp) and mono-nucleosome sized fragments (180–247 bp). (**B**) Asynchronous and mitotic ATAC-seq profiles for an 800 kb region in chromosome 6. (**C**) Comparison of total, short, and mono-nucleosome sized reads for asynchronous and mitotic samples in a 40 kb region in chromosome 6. (**D–F**) Heatmap scatter plots of peak intensities for asynchronous vs mitotic samples in total reads (**D**), short reads (**E**), and mono-nucleosome sized reads (**F**). Linear regression fit and $R^2$ values are shown.

The following figure supplement is available for figure 4:

**Figure supplement 1.** ATAC-seq replicates for asynchronous and mitotic samples.

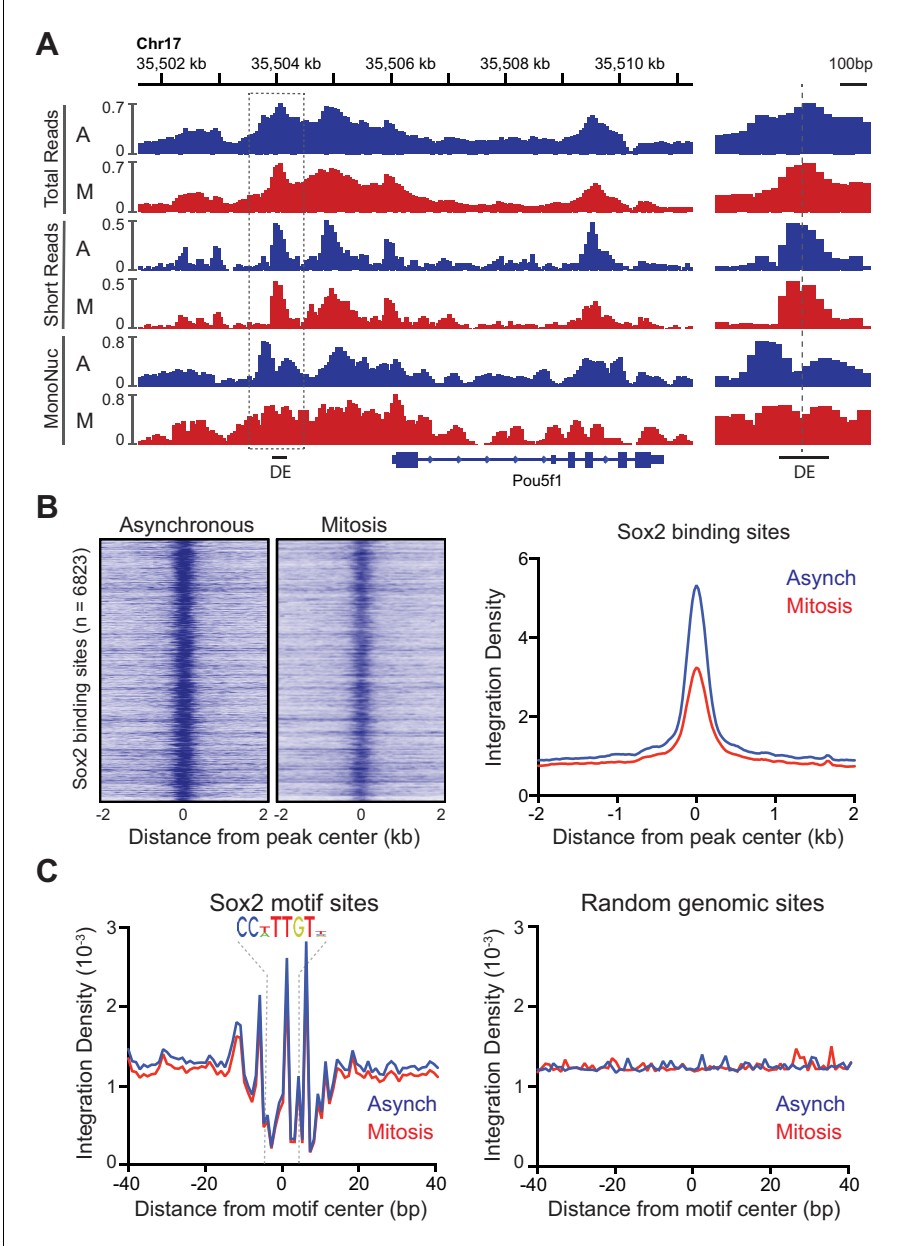

**Figure 5.** Accessibility of Sox2 binding sites in mitosis. (**A**) Comparison of total, short, and mononucleosome sized reads for asynchronous and mitotic samples at the Pou5f1 gene. The boxed region centered at the distal enhancer (DE) is shown in greater detail on the right. (**B**) Heatmaps using the short reads were for asynchronous and mitotic ATAC-seq samples for all Sox2 bound sites (*Chen et al., 2008*) GEO Accession number GSE11431. Right, the average integration density for all Sox2 binding sites is plotted for asynchronous (blue) and mitotic (red) samples. (**C**) Aggregate ATAC-seq footprint for Sox2 (left) and for matched random genomic regions (right).

predicts that the rate of artifact manifestation would increase as a function of PFA concentration. To test this, we performed time-lapse imaging on Halo-Sox2 KI cells while adding varying concentrations of PFA, from 0.25% to 4%, and quantified over time the chromosome enrichment of Halo-Sox2 using H2B-GFP as chromosomal mask. At 60 s post PFA addition, chromosome enrichment of Halo-Sox2 decreased as the concentration of PFA added is increased (*Figure 6B*), confirming that the fixation artifact is dose-dependent. A second prediction based on the model is that TFs with lower $k_{off}$ (more stably bound) would be more resistant to PFA-induced mis-localization. One TF that has been

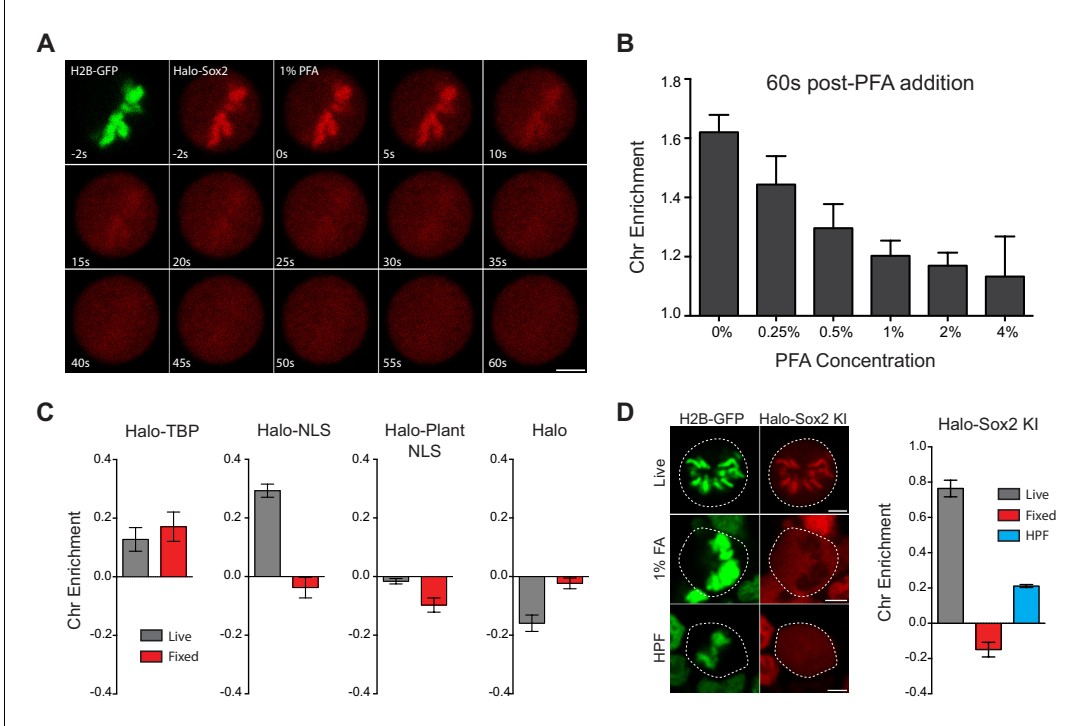

**Figure 6.** Mechanism of formaldehyde-based mis-localization of TFs. (A) Time-lapse two color imaging of endogenously tagged Halo-Sox2 mouse ES cells stably expressing H2B-GFP after adding 1% PFA. (B) Quantification of chromosome enrichment at 60 s after PFA addition with the indicated concentrations of PFA. n = 10 cells. (C) Quantification of chromosome enrichment of indicated HaloTag-fused constructs and HaloTag only in live and fixed conditions. n = 30 cells. (D) High Pressure Freezing and Freeze Substitution was performed on Halo-Sox2 KI cells stably expressing H2B-GFP. Comparison of chromosome enrichment quantification for HPF samples with live and fixed cells are shown. n = 10 cells. Data are represented as mean ± SEM. Scale bars, 5 μm.

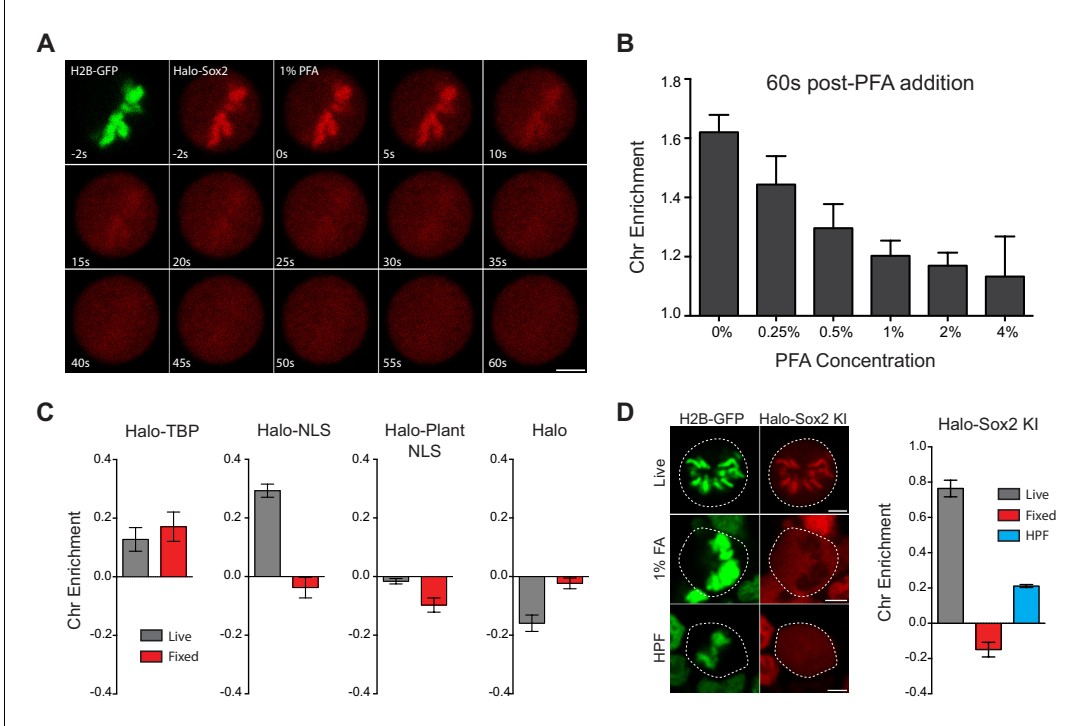

**Video 5.** Time-lapse imaging with PFA addition. Related to *Figure 6*. Mitotic Halo-Sox2 KI cells were imaged at 2 Hz. 1% PFA was added after 10 s of imaging. Movie fps = 20. one pixel = 160 nm.

shown to be stably bound to its target sites is TBP (*Chen et al., 2002*). We imaged Halo-TBP expressing cells before and after fixation and quantified the level of chromosome enrichment as before. Compared to the highly dynamic Halo-NLS, Halo-TBP is resistant to PFA mis-localization (*Figure 6C*). Furthermore, whereas active nuclear import is required for chromosome enrichment, it had no effect on PFA-induced mis-localization as Halo-Plant NLS also becomes excluded from mitotic chromosomes after fixation, in contrast to Halo-only control (*Figure 6C*). A corollary of the second prediction based on the proposed mechanistic model is that if diffusion is reduced while allowing for PFA to equilibrate throughout the cell, the artifact should be reduced. We tested this hypothesis by performing high pressure freezing (HPF) followed by freeze substitution (FS), a method commonly used for preparing samples for electron microscopy. First, the cells are rapidly chilled to liquid nitrogen temperatures in sub-millisecond time scale and under high pressure (2100 bar) to better preserve molecular structures and

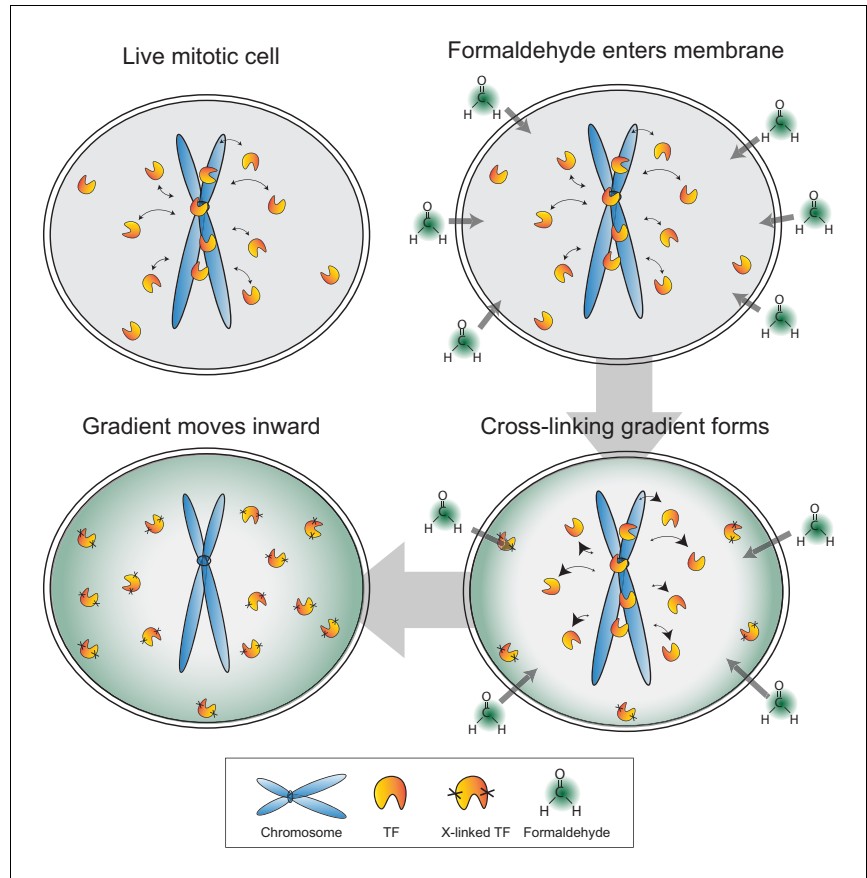

**Figure 7.** Model for formaldehyde-based mis-localization of transcription factors. In live mitotic cells, TFs interact dynamically with mitotic chromosomes with intrinsic $k_{on}$ and $k_{off}$ rates, but can also sample the entire cellular space. During fixation, formaldehyde molecules enter the cell membrane and immediately cross-link with the nearest protein, resulting in a wave of cross-linking gradient that moves from the cell membrane inward. Such a gradient would cross-link cytoplasmic TFs first and result in an effective decrease in $k_{on}$ rates. As the gradient moves to the center, the result is an apparent exclusion of TFs from mitotic chromosomes.

localization. The cells are then immersed in PFA dissolved in an organic solvent at low temperatures over time to allow for a gradual substitution of water molecules with PFA-containing organic solvent. As temperatures warm, the equilibrated PFA molecules begin to crosslink molecules in the cell. We performed this experiment on Halo-Sox2 KI cells labeled with dye immediately before HPF-FS and imaged the cells after HPF-FS (*Figure 6D*). Under these conditions, we partially rescued the localization of Halo-Sox2 on mitotic chromosomes, suggesting that the HPF-FS method can somewhat counteract the PFA-induced artifact. Taken together, these results point to a robust artifact caused by formaldehyde based cross-linking, and support a model whereby highly dynamic TFs are more susceptible to PFA-induced mis-localization.

## Discussion

We initiated our studies by testing whether Sox2 acts as a mitotic bookmarker to maintain the ES cell state. Indeed, we have found that Sox2 is dynamically enriched on mitotic chromosomes and that this interaction is mediated by both the DNA binding domain and the NLS. Furthermore, mitotic chromosomes remain highly accessible, ensuring that TFs such as Sox2 can efficiently and directly sample its binding sites. With these results alone, we might have concluded that Sox2 is one of the few privileged TFs that act as mitotic bookmarkers.

In the process of tracking the behavior of Sox2 during mitosis, however, we unexpectedly discovered that most TFs we sampled continue to associate with mitotic chromosomes in varying levels, in direct contrast to the established literature documenting the eviction of most transcription factors from mitotic chromosomes. Surprisingly, we have found that the previously reported exclusion of TFs from mitotic chromosomes is caused by a chemical crosslinking-based artifact. By combining the dynamic properties of TFs with the chemical properties of formaldehyde, we have described a potential mechanistic model for how the artifact might be produced. Formaldehyde molecules rapidly cross-link to the nearest protein as they cross the cell membrane, resulting in a cross-linking gradient that moves from the cell membrane inward. In mitotic cells, TFs have intrinsic $k_{on}$ and $k_{off}$ rates with respect to mitotic chromosomes, but with the breakdown of the nuclear envelope, can also sample the entire cellular space. The advancing gradient of cross-linking would result in depletion of cytoplasmic TF molecules as they are cross-linked first, reducing the effective $k_{on}$ rates. As the cross-linking gradient moves inward, this effect, combined with the mitotic-specific increase in $k_{off}$, would result in an apparent exclusion of TFs from mitotic chromosomes (*Figure 7*). We have presented evidence that support specific predictions based on this model. In particular, the rate of artifact manifestation is correlated with PFA concentration. Furthermore, highly dynamic TFs are more susceptible to PFA-induced mis-localization compared to stably bound factors. Indeed, some indications of this latter point exist in the literature. For example, factors such as CTCF, the cohesin complex, and DNA topoisomerases, all of which exhibit stable binding, have been shown by fixed immunofluorescence to remain bound on mitotic chromosomes (*Burke et al., 2005*; *Christensen et al., 2002*; *Losada and Hirano, 2001*; *Nakahashi et al., 2013*). The existence of a formaldehyde based artifact, and the potential mode of its action, have several implications. The most direct implication is that it questions decades of research built upon the model that most TFs are evicted and excluded from mitotic chromosomes. Although TFs display varying levels on mitotic chromosomes, from highly enriched to uniform levels, the majority of TFs tested do not exhibit the reported exclusion from mitotic chromosomes, suggesting as yet unknown functions for most TFs during mitosis. The reported TF exclusion has also been implicated in studies of cellular differentiation and nuclear reprogramming (*Egli et al., 2008*), which, in light of our findings reported here, may benefit from reconsideration. A far-reaching implication of the formaldehyde artifact is that many molecular biology techniques aimed at studying dynamics that are dependent on formaldehyde-based fixation may have to be revisited. Chromatin immunoprecipitation (ChIP) studies reliant on formaldehyde cross-linking may underestimate the total binding space of any given TF. Furthermore, studies designed to measure the dynamics of TFs through time-course cross-linked chromatin immunoprecipitation experiments (*Lickwar et al., 2012*) may be susceptible to formaldehyde-based artifacts. Indeed, previous examples of this limitation in ChIP have been documented (*Schmiedeberg et al., 2009*). As technological advances in live-cell and single molecule techniques increase, we can begin to study dynamic processes more accurately.

The visual compaction of chromatin into condensed mitotic chromosomes has led to the general perception that the underlying DNA in mitotic chromosomes is largely inaccessible. However, several experiments point to the idea that DNA remains accessible to some extent in the highly condensed mitotic chromosomes. For example, TFs of the RNA Polymerase I machinery have been shown to rapidly exchange on and off ribosomal DNA clusters during mitosis through FRAP analysis (*Chen et al., 2005*). Diffusion of EGFP molecules is delayed but not hindered through mitotic chromosomes as measured by pair correlation function, suggesting that molecules can freely cross chromosomes during mitosis (*Hinde et al., 2011*). More recently, the Blobel group has performed DNase I hypersensitivity assay coupled with sequencing in mitotic and asynchronous mouse erythroblast cells and has shown that, although there is a global decrease in DNase-seq signals during mitosis, overall patterns of DNase I HS patterns are preserved (*Hsiung et al., 2015*). Our ATAC-seq analysis shows that the levels of integration globally are quantitatively similar for asynchronous and mitotic cells. Taken together, these studies suggest that DNA in mitotic chromosomes is as accessible as in interphase chromatin.

The altered dynamic behavior of Sox2 binding with mitotic chromosomes relative to interphase chromatin implies that TF interactions with target DNA sites are likely cell cycle-dependent. During interphase, Sox2 target search and binding is dominated by three factors: 3D diffusion, protein-DNA contacts, and protein-protein interactions. Our fast-tracking SPT experiments have revealed that, at any given time during interphase, roughly 70% of Sox2 molecules are undergoing diffusion while

30% are bound. Previous studies have shown that bound molecules of Sox2 can be classified further as engaged in fast, non-specific interactions, or stable, site-specific binding (*Chen et al., 2014*). Here, we have confirmed that the interphase long-lived stable binding events are highly dependent on the TAD-mediated protein-protein interactions that occur during transcriptional activation (*Chen et al., 2014*). During mitosis when the nuclear envelope is disassembled and DNA becomes highly condensed, we have observed an increase in the fraction of freely diffusing Sox2 molecules to an average of 82%. Although this greater proportion of freely diffusing molecules could be attributed to an effective increase in total explorable space during mitosis, this effect would have resulted in a cytoplasmic rather than chromosome accumulation of Sox2 during mitosis. Live-cell imaging clearly shows that Sox2 molecules are concentrated at mitotic chromosomes, and that only a small proportion of molecules explore the whole cytoplasmic region. Thus, the majority of Sox2 molecules that are observed on mitotic chromosomes are actually sampling cognate binding sites dynamically. Interestingly, the 18% of Sox2 molecules that are bound on mitotic chromosomes exhibit decreased residence times relative to bound Sox2 molecules in interphase. Indeed, Sox2 dynamics in mitosis mimics the behavior of interphase Sox2 without its transcriptional activation domain, suggesting that Sox2 protein-protein interactions dependent on transcriptional activation are severely abrogated in mitosis. Therefore, in contrast to interphase cells, Sox2 behavior during mitosis appears to be dominated by two main factors, 3D diffusion and protein-DNA contacts. Similarly, FoxA1, another TF that has been described as a mitotic bookmarker, has also been shown to bind to mitotic chromosomes in a more dynamic manner than to interphase chromatin (*Caravaca et al., 2013*). Such cell cycle dependent behavior, as dictated by the changes in global transcriptional status, may be common for many TFs.

We have shown that TF enrichment on mitotic chromosomes is facilitated by a functional NLS, which was surprising given the nuclear envelope breakdown in mitosis. How might the active nuclear import mechanism function to enrich TFs on mitotic chromosomes? During interphase, the canonical nuclear import mechanism employs three main proteins, importins α and β, and the small regulatory GTPase Ran (*Freitas and Cunha, 2009*). Importin α recognizes the NLS-containing protein and binds to importin β to form a cargo complex. Upon import, nuclear Ran-GTP binds to importin β and induces cargo dissociation and the release of the NLS-containing protein. To maintain the directionality of the import mechanism, Ran-GTP must be present at higher concentrations in the nucleus while Ran-GDP must be predominantly cytoplasmic (*Harel and Forbes, 2004*). This compartmentalization is established by the chromatin associated Ran GTP-exchange factor, RCC1, and by the Ran GTPase-activating protein (Ran-GAP), which is localized to the cytoplasmic periphery of the nuclear envelope. Despite nuclear envelope breakdown during mitosis, RCC1 remains localized at mitotic chromosomes, thereby maintaining the Ran-GTP gradient in mitosis (*Moore et al., 2002*). This Ran-GTP gradient has recently been shown to be vital in accurate formation of the spindle assembly during mitosis through the re-purposing of the canonical import proteins (*Forbes et al., 2015*). One possible mechanism for mediating the NLS-dependent enrichment of TFs on mitotic chromosomes is that the maintained Ran-GTP gradient allows importins to retain their ability to transport NLS-containing proteins to mitotic chromosomes. Indeed, the accumulation of NLS containing proteins on chromosomes was hypothesized and modeled previously (*Caudron et al., 2005*) and a drug-dependent block of nuclear import resulted in exclusion of a TF mutant from mitotic chromosomes (*Lerner et al., 2016*). This postulated active shuttling of nuclear proteins to the chromosomes serves two potential functions. First, such a mechanism would facilitate the accumulation of high local concentrations of nuclear factors near the chromosomes to ensure an efficient re-establishment of the nuclear environment immediately following mitosis when the nuclear envelope reforms. Secondly, the increased local concentration of TFs on mitotic chromosomes may contribute to maintaining the accessibility of DNA regulatory regions, the original definition of mitotic bookmarking, even under the highly condensed state of mitotic chromosomes by facilitating increased TF-chromosome interactions. This model is supported by the near perfect concordance of ATAC-seq profiles for asynchronous and mitotic samples, indicating that mitotic chromosomes are as accessible to TFs as interphase chromatin. Therefore, most TFs can and do interact with mitotic chromosomes and thus function as general mitotic bookmarkers.

# Materials and methods

## Cell culture

For all experiments, we used the mouse ES cell line JM8.N4 (RRID: CVCL_J962) obtained from the KOMP repository (https://www.komp.org/pdf.php?cloneID=8669) and tested negative for myco-plasma. ES cells were cultured on gelatin-coated plates in ESC media Knockout D-MEM (Invitrogen, Waltham, MA) with 15% FBS, 0.1 mM MEMnon-essential amino acids, 2 mM GlutaMAX, 0.1 mM 2-mercaptoethanol (Sigma) and 1000 units/ml of ESGRO (Chem- icon). ES cells are fed daily and passaged every two days by trypsinization. Mitotic cells were synchronized by adding 100 ng/ mL of Nocodazole for 6 hr followed by shake-off. Synchronization for other cell cycle stages were performed using the following regime. Cells were incubated in 2 mM Thymidine for 6 hr followed by 6 hr of fresh media. This cycle was repeated twice and cells synchronized in S phase were collected. The double thymidine block was followed by incubation with 100 ng/mL of Nocodazole for 6 hr. Mitotic cells were collected by shake-off, and remaining adherent cells were collected as G2-phase cells. To generate HaloTag-fused TFs, coding sequence of each TF were cloned into a Piggybac vec-tor containing 3X-Flag-Halo-TEV construct upstream of a multiple cloning site. Stably expressing cell-lines were generated by transfecting the Halo-TF Piggybac construct together with the Super Piggybac transposase using Lipofectamine 3000. After 24 hr, stable integration of construct was selected with 1 mg/mL of G418. A similar method was used to generate stable expression of H2B-GFP under puromycin selection at 5 µg/mL.

## Cas9 nuclease plasmid construction

The design of the guide RNA was performed using the CRISPR Design Tool (http://crispr.mit.edu) (*Hsu et al., 2013*). The following oligonucleotide pair (5' CACCGCTGTGGGGGCCGCGCTCG and 5' AAACCGAGCGCGGCCCCCACAGC) was cloned into a modified pX330 vector. The modification involves the inclusion of the Venus coding sequence upstream of the gRNA, driven under the PGK promoter. Venus expression can then be used to sort transfection positive cells. The final vector encoded the gRNA, Venus, and the Cas9 nuclease.

## Design and construction of donor repair vectors

The left homology arm (LHA), HaloTag, and right homology arm (RHA) DNA sequence was con-structed using overlapping PCR. LHA and RHA were amplified from genomic DNA using the follow-ing primer pairs. LHA: 5' GCGCGGTACCGCCAATATTCCGTAGCATGG and 5' CTTTTGCCTTGTTC TCAGCGCTAGCCATGCGGGCGCTGGGCGGGCG. RHA: 5' GAAAGTCCCTGTGTGCGAACCAAG TGGTACGCGTTATAACATGATGGAGACGGAG and 5' GATATCTAGACCTCGGACTTGACCACA-GAG. We introduced silent mutations within the Cas9 nuclease binding region of the right homology arm. The HaloTag was amplified using the following primers to create overlap with LHA and RHA: 5' CGCCCGCCCAGCGCCCGCATGGCTAGCGCTGAGAACAAGGCAAAAG and 5' CTCCGTCTCCA TCATGTTATAACGCGTACCACTTGGTTCGCACACAGGGACTTTC. The final 3-way overlapping PCR was performed with all 3 PCR products as a template and using the following primer pair: 5' GCGCGGTACCGCCAATATTCCGTAGCATGG and 5' GATATCTAGACCTCGGACTTGACCACA-GAG. The final PCR product was cloned into the pUC19 vector.

## Transfection, clone selection, and screening

Mouse ES cells were transfected with of nuclease (0.5 µg) and donor (1 µg) vectors using Lipofect-amine 3000. After 24 hr, cells were trypsinized into single-cell suspension and sorted for Venus expression. Sorted cells were grown on gelatin-coated plates under dilute conditions to obtain indi-vidual clones. After one week, individual clones were isolated and were used for direct cell lysis PCR using Viagen DirectPCR solution (cat #302 C) and the following primers: 5' ACCACGTCCGCTTCA TGGAT and 5' GCCGTAAGGCATCATTGGAC. PCR positive clones were further grown on gelatin-coated plates and cell lysates were obtained for Western blot analysis using α-Sox2 (Millipore Cat# AB5603 RRID:AB_2286686). Three independent homozygous knock-in clones were obtained and fur-ther verified by immunofluorescence using α-Sox2. Halo-Sox2 KI C3 (clone 3) was further tested by teratoma assay and was performed by Applied Stem Cell.

## Live cell imaging and fixed sample immunofluorescence

For all live-cell imaging experiments including FRAP, cells were grown on gelatin-coated glass bottom microwell dishes (MatTek #P35G-1.5–14-C) and labeled with the Halo-ligand dye JF549 at 100 nM concentration for 30 min. Cells were washed 3x with fresh media for 5 min each to remove unbound ligand. Live-cell imaging was performed using a Zeiss LSM 710 confocal microscope equipped with temperature and $CO_2$ control. For fixed imaging, labeled cells were fixed with 4% PFA for 10 min and were washed with 1x PBS prior to imaging. Standard immunofluorescence was performed on labeled cells using 4% PFA. Quantification of chromosome enrichment was performed using Fiji. Epi-fluorescence time lapse imaging was performed on Nikon Biostation IM-Q equipped with a 40x/0.8 NA objective, temperature, humidity and $CO_2$ control, and an external mercury illuminator. Images were collected every 2 min for 12 hr.

## FRAP

FRAP was performed on Zeiss LSM 710 confocal microscope with a 40x/1.3 NA oil-immersion objective and a 561 nm laser. Bleaching was performed using 100% laser power and images were collected at 1 Hz for the indicated time. FRAP data analysis was performed as previously described (*Mueller et al., 2008*). For each cell line, we collected 10 cells for technical replicates in one experiment, which was repeated for a total of three biological replicates (30 cells total).

## Single-molecule microscopy

For all single molecule tracking experiments, indicated cells were grown on gelatin-coated glass bottom microwell dishes (MatTek #P35G-1.5–14-C). For experiments imaging at slow frame rates, cells were labeled with JF549 at 10 pM for 30 min and washed as before. Cells were imaged in ESC media without phenol-red. A total of 600 frames were collected for imaging experiments and were repeated 3x for biological replicates, with each experiment consisting of 10 cells for technical replicates each. Data are represented as mean over experimental replicates (30 cells total) ± SEM. For experiments imaging at fast frame rates, cells were labeled with JF646 at 25 nM as before. Cells were imaged in ESC media without phenol-red. A total of 20,000 frames were collected for fast-tracking imaging experiments and were repeated 3x for biological replicates, with each experiment consisting of eight cells of technical replicates. Data are represented as mean over experimental replicates (24 cells total) ± standard error of means.

## Single particle tracking

Single particle tracking experiments were performed at either a slow frame rate (2 Hz; dye: JF549) to measure residence times or at a high frame-rate (225 Hz; dye: PA-JF646) to measure the displacement distribution and the fraction bound. Imaging experiments were conducted on a custom-built Nikon TI microscope equipped with a 100x/NA 1.49 oil-immersion TIRF objective (Nikon apochromat CFI Apo SR TIRF 100x Oil), EM-CCD camera (Andor iXon Ultra 897), a perfect focusing system (Nikon) and a motorized mirror to achieve HiLo-illumination (*Tokunaga et al., 2008*). To image PA-JF646 dyes (*Grimm et al., 2015*; *2016*), a multi-band dichroic (405 nm/488 nm/561 nm/633 nm BrightLine quad-band bandpass filter, Semrock) was used to reflect a 633 nm laser (1 W, Coherent, Genesis) and 405 nm laser (140 mW, Coherent, Obis) into the objective, and emission light was filtered using a bandpass emission filter (FF01 676/37 Semrock). To image JF549 (*Grimm et al., 2015*), the same multi-band dichroic (405 nm/488 nm/561 nm/633 nm quad-band bandpass filter, Semrock) was used to reflect a 561 nm laser (1 W, Coherent, Genesis) into the objective and emission light was filtered using a bandpass emission filter (Semrock 593/40 nm). The laser intensity was controlled using an acousto-optic transmission filter. For JF549 experiment at 2 Hz, a low constant laser intensity was used to minimize photobleaching. For experiments at 225 Hz, stroboscopic pulses of 1 ms 633 nm laser per frame were used at maximal laser intensity to minimize motion-blurring.

## Single particle tracking: Residence time measurements

When imaging single molecules at long exposure times (500 ms, 2 Hz) fast-moving molecules motion-blur into the background and mostly bound molecules appear as single diffraction limited spots (*Chen et al., 2014*). Thus, bound Sox2 molecules were identified using SLIMfast (*Normanno et al., 2015*) (*Source code 1*), a custom-written MATLAB implementation of

the MTT algorithm (*Sergé et al., 2008*) using the following algorithm settings: Localization error: $10^{-6}$; deflation loops: 1; Blinking (frames); 1; maximum number of competitors: 3; maximal expected diffusion constant (μm²/s): 0.1. The length of each bound trajectory, corresponding to the time before unbinding or photobleaching, was determined and used to generate a survival curve (fraction still bound) as a function of time. The survival curve was then fitted to a two-exponential function:

$$P(t) = F e^{-k_{off,emp,ns}t} + (1-F)e^{-k_{off,emp,s}t}$$

as previously described for Sox2 (*Chen et al., 2014*), where $k_{off,emp,ns}$ corresponds to the empirical off rate for non-specific binding and $k_{off,s}$ corresponds to the empirical off rate for specific binding. The measured empirical off rate is the sum of the photobleaching rate and the actual off rate for unbinding of Sox2 from chromatin:

$$k_{off,emp,s} = k_{photobleach} + k_{off,s}$$

The photobleaching rate was measured using an ES cell line stably expressing H2b-Halo. Since H2b-Halo displays minimal unbinding (e.g. no FRAP recovery), any apparent unbinding was interpreted as photobleaching. Thus, to determine $k_{photobleach}$ the SPT experiment was repeated using identical settings and the apparent off-rate of H2b-Halo determined using two-exponential fitting, where $k_{photobleach}$ corresponds to the slow component. Finally, the Sox2 residence time is then simply given by the inverse of the photobleaching-corrected off rate: $\tau_s = \frac{1}{k_{off,s}}$.

## Single particle tracking: Fraction bound measurements

As with the experiments at 2 Hz, single molecules were localized and tracked using SLIMfast, a custom-written MATLAB implementation of the MTT algorithm (*Sergé et al., 2008*), using the following algorithm settings: Localization error: $10^{-6.25}$; deflation loops: 0; Blinking (frames); 1; maximum number of competitors: 3; maximal expected diffusion constant (μm²/s): 20. To determine the fraction bound from the single particle tracking measurements at 225 Hz, the displacements ('jump lengths') at several Δτ (4.5 ms, 9.0 ms, . . ., 31.5 ms) where fit to a steady-state model consisting of a free Sox2 population (with diffusion constant $D_{FREE}$) and a bound Sox2 population (with diffusion constant $D_{BOUND}$) using a modeling approach similar to what has been previously described by *Mazza et al. (2012)*, but with some modifications. The combined displacement histograms at several Δτ (4.5 ms, 9.0 ms, . . ., 31.5 ms) were fitted to:

$$P(r,\Delta\tau) = F_{BOUND}\frac{r}{2(D_{BOUND}\Delta\tau+\sigma^2)}e^{\frac{r^2}{4(D_{BOUND}\Delta\tau+\sigma^2)}} + Z_{CORR}(\Delta\tau)(1-F_{BOUND})\frac{r}{2(D_{FREE}\Delta\tau+\sigma^2)}e^{\frac{r^2}{4(D_{FREE}\Delta\tau+\sigma^2)}}$$

where:

$$Z_{CORR}(\Delta\tau) = \frac{1}{\Delta z}\int_{-\Delta z/2}^{\Delta z/2}\left\{1 - \sum_{n=0}^{\infty}(-1)^n\left[erfc\left(\frac{\frac{(2n+1)\Delta z}{2}-z}{\sqrt{4D_{FREE}\Delta\tau}}\right) + erfc\left(\frac{\frac{(2n+1)\Delta z}{2}+z}{\sqrt{4D_{FREE}\Delta\tau}}\right)\right]\right\}dz$$

and:

$$\Delta z = 0.700~\mu m + 0.15716 s^{-1/2}\sqrt{D} + 0.20811~\mu m$$

using custom-written least-squares fitting software (MATLAB). The above model contains three fitted parameters ($D_{BOUND}$, $D_{FREE}$ and $F_{BOUND}$). The localization error, σ, was roughly 35 nm. $Z_{CORR}(\Delta\tau)$ corrects for free molecules moving out of the axial detection slice (out-of-focus; axial detection slice experimentally measured to be ~0.7 μm). The $Z_{CORR}(\Delta\tau)$ expression assumes absorbing boundaries, which overestimates the fraction of molecules moving out of focus (*Kues and Kubitscheck, 2002*). To correct of this, Monte Carlo simulations were performed to determine a corrected Δz, which is shown above. Full details on modeling displacements will be published elsewhere (Hansen *et al.*) and is available upon request.

## ATAC-seq

ATAC-seq was performed on 50,000 asynchronous and mitotic cells as previously described (*Buenrostro et al., 2013*) with the following modifications. For both asynchronous and mitotic samples, Nocodazole was added to cells at 100 ng/mL concentration and incubated at 37 C for 6 hr, and mitotic cells were collected by careful shake off. After PBS wash, cells were immediately resuspended in the transposase reaction mix (25 µL 2x TD buffer, 2.5 µL transposase, and 22.5 µL nuclease-free water). Transposition and DNA purification was performed as described (*Buenrostro et al., 2013*). For PCR amplification, we determined the linear range to be between 10–16 cycles and performed amplification for library preparation using 12 cycles for all subsequent samples. We performed two replicates, and each sample and replicate was sequenced using one lane of Illumina Hi-Seq 2500 for 50 bp paired-end reads.

## ATAC-seq data analysis

Sequenced paired mates were mapped on mm10 genome build using Bowtie2 with the following parameters: –no-unal –local –very-sensitive-local –no-discordant –no-mixed –contain –overlap –dovetail –phred33 –I 10 –X 2000. Raw mapped reads were visualized using Integrative Genomics Viewer (IGV). From the length distribution of sequenced reads, two size classes were used: short reads (under 100 bp) and mono-nucleosome sized reads (180–247 bp). Fragment sizes corresponding to these size classes were mapped separately using the same parameters as above. Peak calling was performed using the Homer suite package for asynchronous and mitotic samples, and using either total reads, or each size class. Peaks for asynchronous and mitotic samples were merged for the corresponding size classes. Peak intensity scatter heatmap was plotted using the MATLAB package Heatscatter with the number of bins set at 100. Sox2 binding sites were collated from published ChIP-seq data sets (*Chen et al., 2008*) with GEO Accession number GSE11431. ATAC-seq read densities from the short reads size class were calculated in a 4 kb region centered at each peak in 25 bp bins were calculated using a Homer suite package (*Heinz et al., 2010*), and visualized using Java TreeView (*Saldanha, 2004*). Sox2 motif and associated position weight matrix were obtained from JASPAR core 2014 database, and genomic locations containing Sox2 motif was generated using PWMTools with a cut-off p-value of $10^{-4}$. The short read size class was used to calculate the read density at single base resolution for footprinting analysis. Footprinting was performed similarly for matched random genomic regions. Sequencing data are deposited into GEO under the accession number GSE85184.

## Time lapse imaging—quantification of PFA movies

Halo-Sox2 KI cells stably expressing H2B-GFP were grown on gelatin-coated chambered coverglass (Lab-Tek #155411), labeled with JF-549 at 100 nM, and washed as described above. Time lapse imaging was performed with Zeiss LSM 710 confocal microscope as described above, with images collected at 2 Hz for 2.5 min. PFA at varying concentrations were added 10 s after the start of image collection. Quantification of chromosome enrichment was performed using an in-house Matlab code (*Source code 2*). Briefly, for each frame, the H2B-GFP was thresholded to generate a mask for chromosomes. The mask was applied to the JF549 channel to calculate the mean intensity at the chromosomes, which was normalized to the total cell intensity. The normalized chromosome intensity at 60 s post PFA-addition was obtained for each experiment and plotted as mean ± SEM.

## Acknowledgements

We thank Gina M Dailey for technical assistance with cloning and Astou Tangara for microscope assembly and maintenance. We thank the following persons and shared facilities: Kartoosh Heydari and the flow cytometry core facility; Shana L McDevit and the QB3 Vincent J Coates Genomics Sequencing Laboratory; Kent L McDonald and the Electron Microscope Lab. This work was performed in part at the CRL Molecular Imaging Center, supported by the Gordon and Betty Moore Foundation. This work used the Vincent J Coates Genomics Sequencing Laboratory at UC Berkeley, supported by NIH S10 Instrumentation Grants S10RR029668 and S10RR027303. This investigation has been aided by a grant from the Jane Coffin Childs Memorial Fund for Medical Research (SST).

This work was supported by NIH grant UO1-EB021236 (XD), by the California Institute for Regenerative Medicine grant LA1-08013 (XD and RT), and by the Howard Hughes Medical Institute (RT).

## Additional information

### Competing interests
RT: President of the Howard Hughes Medical Institute (2009-present), one of the three founding funders of eLife, and a member of eLife's Board of Directors. The other authors declare that no competing interests exist.

### Funding

| Funder | Author |
| --- | --- |
| Howard Hughes Medical Institute | Robert Tjian |
| Jane Coffin Childs Memorial Fund for Medical Research | Sheila S Teves |

The funders had no role in study design, data collection and interpretation, or the decision to submit the work for publication.

### Author contributions
SST, Conception and design, Acquisition of data, Analysis and interpretation of data, Drafting or revising the article; LA, Acquisition of data, Analysis and interpretation of data; ASH, Analysis and interpretation of data, Drafting or revising the article; LX, Drafting or revising the article, Contributed unpublished essential data or reagents; XD, RT, Conception and design, Drafting or revising the article

### Author ORCIDs

Sheila S Teves, http://orcid.org/0000-0002-1220-2414
Robert Tjian, http://orcid.org/0000-0003-0539-8217

## Additional files

### Supplementary files
• Source code 1. Slimfast.

• Source code 2. Mitotic chromosome enrichment.

### Major datasets
The following dataset was generated:

| Author(s) | Year | Dataset title | Dataset URL | Database, license, and accessibility information |
| --- | --- | --- | --- | --- |
| Teves SS, Tjian R | 2016 | Global accessibility of mitotic chromosomes | https://www.ncbi.nlm.nih.gov/geo/query/acc.cgi?acc=GSE85184 | Publicly available at the NCBI Gene Expression Omnibus (accession no: GSE85184) |

The following previously published dataset was used:

| Author(s) | Year | Dataset title | Dataset URL | Database, license, and accessibility information |
|---|---|---|---|---|
| Wei C | 2008 | Mapping of transcription factor binding sites in mouse embryonic stem cells | https://www.ncbi.nlm.nih.gov/geo/query/acc.cgi?acc=GSE11431 | Publicly available at the NCBI Gene Expression Omnibus (accession no: GSE11431) |

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
