## [Decision Letter]

Thank you for submitting your article "A Dynamic Mode of Mitotic Bookmarking by Transcription Factors" for consideration by *eLife*. Your article has been favorably evaluated by Jessica Tyler (Senior Editor) and three reviewers, one of whom is a member of our Board of Reviewing Editors.

The reviewers have discussed the reviews with one another and the Reviewing Editor has drafted this decision to help you prepare a revised submission.

Summary:

This very nice manuscript from the Tjian laboratory clearly has the impact and level of interest to merit publication in *eLife*. In this work, the authors address the nature of mitotic bookmarking, and the traditional presumption that condensation of mitotic chromatin renders it inaccessible to binding by transcription factors. This view has been changing recently, and this work represents an important step towards defining how transcriptional memory is retained across mitosis and the ways in which protein-DNA interactions are altered through the cell cycle.

The primary biological findings in this work are: i) that *Sox2*, Oct4 and Esrrb, all factors important for maintenance of the ES cell state, are able to interact with mitotic chromatin, and ii) that the overall accessibility of mitotic chromatin is only modestly lower than during interphase. Further, the authors reveal and explore a technical artifact that has been noted over the years, but never fully understood: they show that formaldehyde cross-linking creates the appearance of exclusion of transcription factors from mitotic chromatin, likely due to the fast k_off_ of factors from mitotic DNA and trapping of dissociated factors in the nucleoplasm as formaldehyde crosslinks interfere with positive DNA-binding surfaces and/or nuclear localization signals. This fixation artifact leads to the apparent exclusion from mitotic chromatin of factors such as Sp1, Klf4, Foxo1and Foxo3a that are evenly distributed through the nucleoplasm in live mitotic cells. Thus, the authors conclude that the use of formaldehyde crosslinking has perpetuated the notion that mitotic chromatin is unavailable for transcription factor binding, when in fact, factors are fully able to explore their binding sites. Finally, the authors propose that it is the absence of stabilizing interactions with the transcription machinery, rather than an inherent decrease in accessibility that underlies the reduced residence time of transcription factors on mitotic chromatin.

Essential revisions:

Most of the data are quite convincing and the findings are very significant for our understanding of gene regulation. However, there remain some issues that the authors should clarify through additional experimentation and through further discussion in the text.

1) Even in live cell imaging experiments, most TFs investigated do not show any enrichment on mitotic chromatin, and those that do are lineage-specifying factors. Transcription factors Sp1 and Foxo have no apparent enrichment on mitotic chromatin, and HSF1 and Stat3 are considered 'de-enriched'. In this way, the current manuscript is in agreement with a good deal of previously published literature showing that pioneer, or lineage-specifying factors have a greater propensity to interact with mitotic chromatin that other factors. However, the authors seem to be arguing against this idea in the discussion, which is confusing. In addition, it isn't clear how significant the level of binding observed for factors like Sp1 is as compared to factors that are truly not associated with mitotic chromatin.

Thus, we suggest two improvements to Figure 1 and its discussion:i) Figure 1 should be accompanied by negative controls for proteins that should not be present on mitotic chromosomes to assess the level of contamination of the mitotic chromatin by chromatin from other cell cycle stages and the level of background in the assay. Cell synchronization was done using nocodazole and shake off, but mouse cells are known to have a leaky response to nocodazole and so some will likely slip through the block during the 6 hours. The same applies to Figure 1 – +ve and -ve controls for the cytoplasmic and chromatin fractions need to be included in the immunoblotting.ii) The authors should clarify in the Results and Discussion that even using live cell techniques, not all factors interact in the same manner with mitotic chromatin, and that their findings in this regard are generally in line with previous work indicating a preference for lineage-specifying, 'pioneer' factors to be enriched on chromatin during mitosis.

2) The similarity in the pattern of ATAC-sequencing data between mitotic and interphase cells is potentially very significant, since it is the best indication that the TFs bind their correct recognition sites in mitotic chromosomes rather than just being non-specifically associated. However, much of the discussion of the ATAC-seq does not match the data presented, which is confusing. First and foremost, in order to quantitatively compare accessibility between two different conditions that may have dramatically different chromatin structure, the ATAC-seq experiments must include an internal control (e.g. spike-in with nuclei of another species, such as *Drosophila*).

Also, the number of cells used per reaction should be stated. Given the contamination of the mitotic fraction with some interphase cells, there is some concern that the ATAC-seq signals are coming from this – albeit small – fraction of contaminating interphase cells rather than from the mitotic chromosomes.

---

## [Author Response]

[…]

*Essential revisions:*

*Most of the data are quite convincing and the findings are very significant for our understanding of gene regulation. However, there remain some issues that the authors should clarify through additional experimentation and through further discussion in the text.*

*1) Even in live cell imaging experiments, most TFs investigated do not show any enrichment on mitotic chromatin, and those that do are lineage-specifying factors. Transcription factors Sp1 and Foxo have no apparent enrichment on mitotic chromatin, and HSF1 and Stat3 are considered 'de-enriched'. In this way, the current manuscript is in agreement with a good deal of previously published literature showing that pioneer, or lineage-specifying factors have a greater propensity to interact with mitotic chromatin that other factors. However, the authors seem to be arguing against this idea in the discussion, which is confusing. In addition, it isn't clear how significant the level of binding observed for factors like Sp1 is as compared to factors that are truly not associated with mitotic chromatin.*

*Thus, we suggest two improvements to Figure 1 and its discussion:i) Figure 1 should be accompanied by negative controls for proteins that should not be present on mitotic chromosomes to assess the level of contamination of the mitotic chromatin by chromatin from other cell cycle stages and the level of background in the assay. Cell synchronization was done using nocodazole and shake off, but mouse cells are known to have a leaky response to nocodazole and so some will likely slip through the block during the 6 hours. The same applies to Figure 1 – +ve and -ve controls for the cytoplasmic and chromatin fractions need to be included in the immunoblotting.*

We agree and we originally performed the fractionation experiments to include TBP (which had previously been shown to interact with mitotic chromosomes) and Pol II as the negative control. However, because the current *Sox2* story has diverged and encompasses its own complete study, we are also preparing a manuscript on TBP and the general transcriptional machinery during mitosis where we plan to include the immunoblots for TBP and Pol II. If the reviewers feel that this data is critical for the current manuscript, we will include it in Figure 1—figure supplement 2.

*ii) The authors should clarify in the Results and Discussion that even using live cell techniques, not all factors interact in the same manner with mitotic chromatin, and that their findings in this regard are generally in line with previous work indicating a preference for lineage-specifying, 'pioneer' factors to be enriched on chromatin during mitosis.*

In our findings, we primarily distinguished between exclusion and non-exclusion, which includes all the varying levels of enrichment, from uniform to highly enriched. We used this distinction because mitotic bookmarking depended on the fact that most TFs are excluded. We agree that the level of enrichment in live cells may distinguish lineage specific factors from other TFs, but if bookmarking is defined by binding alone, we cannot rule out that the lower levels of enrichment of non-lineage specific factors (e.g. Foxo and Sp1) also have a functional effect in maintaining transcription programs through mitosis. In the Results section, we describe that “the majority of these factors exhibited varying protein levels on mitotic chromosomes, from highly enriched to uniform levels.” To clarify our conclusions in the Discussion section, we have modified the part that discusses this result to say the following: “Although TFs display varying levels on mitotic chromosomes, from highly enriched to uniform levels, the majority of TFs tested do not exhibit the reported exclusion from mitotic chromosomes, suggesting as yet unknown functions for most TFs during mitosis.”

*2) The similarity in the pattern of ATAC-sequencing data between mitotic and interphase cells is potentially very significant, since it is the best indication that the TFs bind their correct recognition sites in mitotic chromosomes rather than just being non-specifically associated. However, much of the discussion of the ATAC-seq does not match the data presented, which is confusing. First and foremost, in order to quantitatively compare accessibility between two different conditions that may have dramatically different chromatin structure, the ATAC-seq experiments must include an internal control (e.g. spike-in with nuclei of another species, such as Drosophila).*

*Also, the number of cells used per reaction should be stated. Given the contamination of the mitotic fraction with some interphase cells, there is some concern that the ATAC-seq signals are coming from this – albeit small – fraction of contaminating interphase cells rather than from the mitotic chromosomes.*

Our request to the editor for clarification:

Thank you for the very rapid turn-around time for the review and the many helpful and insightful comments. As we prepare our response letter and revised manuscript, we wanted to clarify one aspect regarding point 2 in the "Essential Revisions" section of the decision letter. We are a little perplexed regarding the recommendation to repeat the ATAC-seq experiments with spike-in controls. To our knowledge, spike-in controls are useful and recommended when comparing two samples where the quantity of the starting material is unknown, such as in typical RNA-seq analyses. However, for our experiments, we performed the ATAC-seq with carefully counted aliquots containing 50,000 cells for each asynchronous and mitotic population, as outlined in the Buenrostro et al. study. This allowed us to know precisely the quantity of the starting material in comparing samples, and so we reasoned that the resulting sequencing reads would be quantitatively comparable. Indeed, to our knowledge, ATAC-seq experiments in the growing published literature have routinely and generally been performed without spike-in controls for this reason.

Before we launch into redoing these rather expensive and time consuming experiments, we wanted to clarify this issue and wondered if the reviewers have some specific reason why they think spike-in controls are necessary in our case and believe that redoing the ATAC-seq with spike-in controls is critical to support our conclusions as outlined in the manuscript.

Response from the Editor

*Thank you for your enquiry. The reviewing editor and I have discussed this further, and we would like further clarification as to how your ATAC-seq was performed to be included in your Methods section. Specifically, 1. Was the PCR amplification of the adapter ligated DNA fragments in the linear/exponential range for all samples? 2. Was the sequencing set to read approximately the same number of reads for each sample? 3. Was more than 50% of the mitotic cells verified to be arrested in mitosis?*

*If the answers are 1. yes and 2. no and 3. yes respectively, then we would agree with your conclusions on the ATAC-seq.*

*If the answer to any of these three questions is different, then we would request that you either repeat the assay with spike-in controls (at least if the answers to question 1 and 2 are different) or acknowledge in the text that the ATAC-seq analysis does not address relative global accessibility in mitosis vs interphase due to lack of normalization controls / due to contamination of interphase cells.*

*I am sure that you can appreciate the need for the readers to know this information, given the increasing awareness of the artifacts that can creep into this sort of genomic assay.*

Our response to the Editor:

Thanks so much for the quick response. We agree that there is definitely a high bar that needs to be met to avoid artifacts, especially for genomic assays.

For question 1, yes the PCR amplification was in the linear range. We performed the qPCR test for linear range as recommended in the original ATAC-seq protocol and determined that our samples are in the linear range within 10-16 cycles of amplification. We chose 12 cycles for all ATAC-seq amplifications and kept it consistent for asynchronous and mitotic samples. This is now included in the Methods section (subsection “ATAC-seq”). We have also included the number of cells per ATAC-seq experiment.

For question 2, no the sequencing was not set to read the same number of reads for each sample. We have reported the statistics for ATAC-seq libraries in Table 1 currently within Figure 4—figure supplement 1. Fortuitously, the number of reads across samples and replicates are fairly similar. For all the analyses, the reads are normalized for total number of sequenced reads per sample and replicate.

For question 3, yes, the mitotic cells are verified to be of ~95% pure population, both by immunofluorescence, and by imaging cells expressing H2B-GFP. We have also performed a time course imaging experiment during the ATAC-seq incubation protocol using cells expressing H2B-GFP to visualize the condensed chromosomes, and this showed that the mitotic cells remain arrested at mitosis throughout the tagmentation part of the experiment. We plan to include this data in the figure supplement.

We have clarified and added these experimental details in the manuscript as stated above.